# The relation of fibromyalgia and fibromyalgia symptoms to self-reported seizures

Johannes J. Rasker[1], Frederick Wolfe[2,3], Ewa G. Klaver-Krol[4], Machiel J. Zwarts[5], Peter M. ten Klooster[1] *

1 Faculty of Behavioral Management & Social sciences, Department of Psychology, Health & Technology, University of Twente, Enschede, The Netherlands, 2 National Data Bank for Rheumatic Diseases, Wichita, Kansas, United States of America, 3 University of Kansas School of Medicine, Wichita, Kansas, United States of America, 4 Roessingh Research and Development, Enschede, The Netherlands, 5 Epilepsy Center Kempenhaeghe, Heeze, The Netherlands

* p.m.tenklooster@utwente.nl

**Data Availability Statement:** All relevant data are within the manuscript and its Supporting Information files.

## Abstract

### Objective

Several epidemiological and clinical reports associate fibromyalgia (FM) with seizure disorders, and clinical studies associate FM diagnosis with psychogenic non-epileptic seizures. However, these associations rely on self-reports of being diagnosed with FM or unstandardized clinical diagnosis in combination with small samples. We investigated the association of FM and self-reported seizures using a large rheumatic disease databank and the current established self-reported, symptom-based FM diagnostic criteria.

### Methods

We selected a random observation from 11,378 subjects with rheumatoid arthritis (RA), 2,390 (21.0%) of whom satisfied 2016 revised criteria for FM. Patients were inquired about the presence of any kind of seizures in the previous 6 months, anti-epileptic medications, and patient-reported symptoms and outcomes.

### Results

Seizures were reported by 89 RA patients who met FM criteria (FM+) and by 97 patients who did not (FM-), resulting in an age- and sex-adjusted seizure prevalence of 3.74 (95% CI 2.95 to 4.53) per 100 FM+ subjects and 1.08 (95% CI 0.87 to 1.30) in FM- subjects. The seizure odds ratio of FM+ to FM- cases was 3.54 (95% CI 2.65 to 4.74). Seizures were associated to a very similar degree with symptom reporting (somatic symptom count and comorbidity index) as to FM diagnosis variables. RA patients reporting seizures also reported worse pain, quality of life, and functional status. Seizure patients treated with anti-seizure medication had worse outcomes and more comorbidities than seizure patients with no seizure drugs.

### Conclusions

We found a significant and similar association of both FM diagnostic variables and FM-related symptom variables, including the number of symptoms and comorbidities, with self-

**Funding:** This research did not receive any specific grant from funding agencies in the public, commercial, or not-for-profit sectors.

**Competing interests:** The authors have declared that no competing interests exist.

reported seizures in people with RA. The observed association was similar to those found in previous studies of symptoms variables and seizures and does not suggest a unique role for fibromyalgia diagnosis. Rather, it suggests that multi-symptom comorbidity is linked to seizures in a complex and not yet clearly understood way. As the current study relied on self-reported seizures and was not able to distinguish between epileptic and psychogenic none-pileptic seizures, future studies are needed to replicate the findings using both validated FM criteria assessments and clinically verified diagnoses of epileptic and psychogenic seizures.

## Introduction

The concept of fibromyalgia (FM) achieved prominence with the publication of the American College of Rheumatology criteria for fibromyalgia in 1990 [1]. The first detailed studies of a fibromyalgia-seizure disorder association occurred in 2005 when population studies identified an increased prevalence of FM in subjects with epilepsy [2] and clinical studies found a strong association between psychogenic non-epileptic seizures (PNES) and FM [3]. Since then, several epidemiologic and clinical studies have also noted this association and some have addressed possible causes [2–11].

FM is a much-debated and controversial condition. While the diagnostic term is endorsed by many physicians and institutions, others have questioned whether fibromyalgia is a real disease or disorder [12–14]. Also, some physicians and researchers prefer to address the disorder with alternative names, such as somatic symptom disorder or various regional pain disorders [15]. As a result, studies have suggested that FM is often both under- and overdiagnosed. For instance, as many as 3 out of 4 people who could receive the diagnosis of FM are estimated to remain undiagnosed [16]. On the other hand, a recent study by Walitt et al. [17] indicated that almost 75% of people who self-report a physician diagnosis of FM do not satisfy FM criteria, suggesting that clinical diagnosis often tends to be off-hand and by gestalt [18, 19].

These controversies and diagnostic problems may also be germane to the reported associations between FM and seizures, because all published epidemiologic studies of FM and seizures have relied on self-report clinical diagnosis of fibromyalgia. In addition, clinical studies that utilized patients with fibromyalgia did not apply FM criteria or applied these criteria non-uniformly to patients and controls. Such problems may lead to unreliable estimates of both prevalence and disease associations.

Starting in 1998, we collected semiannual data of patients with rheumatic diseases as part of an ongoing longitudinal databank [20, 21]. Beginning in 2010, we administered the newly developed self-report FM criteria [22–24], which are based on widespread pain and a mix of other common symptoms instead of the previous tender point rule [1], to all participating rheumatologic patients. In this study, we used these criteria to examine the relationship of both FM diagnostic variables and other FM-related symptoms and comorbidities with self-reported seizures in order to extend the knowledge about the relationship between FM and seizures and to address the diagnostic problems raised in prior publications. Additionally, we explored associations between self-reported seizures and patient-reported outcomes of pain, quality of life, and functional status.

## Methods

### Patient selection

We used the longitudinal research database of the National Data Bank for Rheumatic Diseases (NDB) to evaluate the presence of any kind of seizures and FM-related issues in the included

rheumatoid arthritis (RA) patients. The methodology of the NDB has been reported previously [20, 21]. Briefly, the databank consists of more than 50,000 patients with various rheumatic diseases, including RA, osteoarthritis, lupus, and FM under the care of more than 1,500 rheumatologists from every region of every US state and Canadian province. Participating patients are surveyed with mailed and online comprehensive survey questionnaires every 6 months. Beginning in 2010, the NDB added FM criteria items to the questionnaire consistent with the then new American College of Rheumatology (ACR) 2010 preliminary criteria and current 2016 revised criteria for the diagnosis of FM [22, 24]. The full comprehensive 27-page questionnaire can be downloaded from: https://www.arthritis-research.org/research/forward-research-documents.

For the current study, we specifically selected all RA patients who were referred to the NDB only because they had RA and who had completed the new FM criteria at least once to provide a base population unselected for any symptom. They had not been evaluated in any way for the presence of FM and were not selected in any way for FM characteristics. For analysis, we selected a random observation from 11,378 RA participants, consisting of 9,303 (81.8%) women and 2,075 (18.2%) men (S1 Data and S1 Text). The mean age (SD) of participants at the time of the observation was 61.0 (13.4) years. Using an RA disease sample has the added advantage of being enriched by fibromyalgia cases, since the prevalence rate of comorbid FM in patients with inflammatory arthritis is considerably higher than among the general population with pooled prevalence estimates of 18–24% in RA [25]. A detailed description of the RA dataset used in this study is available elsewhere [26]. Data capture rates in the NDB are high. In general, our comprehensive questionnaires and electronic equivalents have been completed at 92.7% of observations, and had missing data rates for common patient-reported variables such as the Health Assessment Questionnaire (HAQ) and SF-36 of 0.4% and 3.2%, respectively [27].

## Measures

Participating RA patients self-completed a battery of questionnaires, including the ACR FM diagnostic criteria questionnaire [23], clinical and treatment characteristics (e.g., medication use, comorbidities) and other key rheumatic disease outcomes.

**FM diagnostic variables and criteria.**   The definition of a positive FM case (FM+) was based on the 2016 revision of the ACR 2010 criteria [24]. The criteria are based on the Widespread Pain Index (WPI) and the Symptom Severity Scale (SSS) [23], in combination with complying to a set definition of generalized pain and duration of symptoms.

The WPI, with a score of 0–19, is a summary count of the number of 19 painful regions from the Regional Pain Scale (RPS), a self-reported list of painful regions [28]. The SSS, with a score of 0–12, is the sum (0–9) of the severity scores of 3 symptoms (fatigue, waking unrefreshed, and cognitive symptoms) plus the sum (0–3) of the following three symptoms the patient has been bothered by during the previous 6 months: headaches (0–1), pain or cramps in lower abdomen (0–1) and depression (0–1).

The notion of widespread pain utilizes the definition in the 1990 FM criteria [1] that "Pain is considered widespread when all of the following are present: pain in the left side of the body, pain in the right side of the body, pain above the waist, and pain below the waist. In addition, axial skeletal pain (cervical spine or anterior chest or thoracic spine or low back) must be present. For the notion of generalized pain (added to the revised 2016 criteria), pain in at least 4 of 5 regions must be present [24]. Jaw, chest, and abdominal pain are not included in the generalized pain definition. A patient satisfies the revised 2016 criteria if the following 3 conditions are met: (1) WPI $\geq$ 7 and SSS $\geq$ 5 or WPI is 4–6 and SSS $\geq$ 9, (2) generalized pain is present, and (3) symptoms need to have been generally present for at least 3 months [24].

Besides their use for diagnosing FM, the WPI and SSS can be summed into a continuous poly-symptomatic distress (PSD, range 0–31) scale, also known as the Fibromyalgia Severity score (FS). The PSD is a measure of "fibromyalgianess" and indicates the magnitude and severity of fibromyalgia symptoms in those satisfying and not satisfying FM criteria [29]. The continuous PSD score has been shown to more usefully identify and predict disproportionate responses in RA patients.

**FM-related variables.** As fibromyalgia is defined by symptom presence and intensity, it can be characterized and measured by a number of scales in addition to meeting vs. not meeting the criteria, both variables that are part of the formal diagnostic criteria (WPI, SSS and PSD) and other FM-related variables not included in the FM criteria, such as a total symptom count and the number of comorbidities [30, 31]. A total symptom count was calculated for this study based on a checklist of 60 specific symptom, divided over 8 categories (musculoskeletal, gastrointestinal tract, skin, blood, head / eyes / ears / nose / mouth / throat, neurological and psychological, chest / lungs / heart, urine and kidneys). We additionally calculated a patient-reported composite comorbidity index (range 0–11) that was composed of 11 present or past comorbid conditions, including pulmonary disorders, myocardial infarction, other cardiovascular disorders, stroke, hypertension, diabetes mellitus, spine/hip/leg fracture, gastrointestinal (GI) ulcer, other GI disorders, cancer, and depression [32].

**Seizures and seizure-related medication.** We determined the presence or absence of self-reported seizures by the patient's selecting the response option "Seizures or convulsions" in response to the question: "During the past 6 months have you had any of the following symptoms?". We also determined if patients had reported the use of any of the seizure drugs listed by Epilepsy Foundation (https://www.epilepsy.com/learn/treating-seizures-and-epilepsy/seizure-medication-list) and the duration of this therapy in the previous 6 months. The anticonvulsant pregabalin has also been approved by the US Food and Drug Administration (FDA) to treat the neural pain associated with FM.

**Patient-reported outcome measures.** Participants reported their functional status over the past week using the Health Assessment Questionnaire (HAQ, range 0–3) with higher scores indicating more disability [33]. They also completed a visual analog scale for pain over the past week that was scored as 0–10 with anchors ranging from no pain to severe pain. Health-related quality of life was measured using the EuroQol (EQ-5D) health thermometer [34]. A score of 100 indicates perfect health.

## Statistical analyses

Data were analyzed using Stata version 15.0 [35]. Associations of fibromyalgia-related variables and seizure treatment drugs with self-reported seizures and the seizure treatment drugs in the past 6 months were analyzed with logistic regressions and expressed as crude odds ratios (ORs) with 95% confidence intervals (CIs). The prevalence rate and OR for patients which met the 2016 FM criteria (FM+) were additionally adjusted for age and sex. The discriminative value of FM diagnostic variables (WPI, SSS and PSD) and other FM-related variables (symptom count and comorbidity index) for seizures was compared by computing their areas under the receiver operating characteristics curves (ROC AUC) with 95% CIs. Differences in ROC AUCs were tested using the method by Delong et al. [36]. Differences in patient characteristics and patient-reported outcomes between RA patients with and without seizures and between seizure patients with and without seizure drugs were analyses descriptively.

## Ethics

This study was conducted in accordance with the ethical standards of the responsible committee on human experimentation and with the Helsinki Declaration of 1975, as revised in 1983.

Participants in the NDB are volunteers who agree to complete questionnaires at 6-month intervals. Each patient completes a written consent that is part of the larger 27 page questionnaire. The Via Christi Institutional Review Board (Wichita, Kansas, USA) has reviewed and approved the ongoing data collection and methods of the NDB with respect to long-term outcomes in rheumatic diseases, specific individual studies are not required to be approved. All data for this study was completely anonymized and de-identified before analysis.

## Results

### Association between FM diagnosis and seizures

Of the 11,378 subjects with rheumatoid arthritis, 2,390 (21.0%) satisfied the revised 2016 FM criteria (Table 1). In the total sample of all RA patients, seizures were reported by 186 patients, resulting in an overall adjusted seizure prevalence of 1.63 (95% CI 1.40 to 1.87) per 100 subjects. Eighty-nine FM+ patients (3.72%) and 97 FM- patients (1.08%) reported seizures, resulting in a substantially higher adjusted seizure prevalence rate of 3.74 (95% CI 2.95 to 4.53) per 100 FM+ patients compared to 1.08 (95% CI 0.87 to 1.30) in FM- patients. The seizure odds ratio of FM+ to FM- patients was 3.54 (95% CI 2.97 to 4.42).

Although 9,303 RA patients (81.7%) were women, corresponding with the predominance of women in RA, women represented only 57.8% of those reporting seizures and 61.4% of those satisfying FM criteria. This indicates that the association of female gender with both seizures and fibromyalgia was limited in the present sample.

### Association of FM diagnostic and FM-related variables with seizures

The association of individual FM diagnostic variables with seizures was supported for all three variables (PSD, WPI and SSS; Table 2). The total polysymptomatic distress scale (PSD) score, which models fibromyalgia as a continuum disorder, was most strongly associated with self-reported seizures with a ROC AUC of 0.683 (Table 2). However, its discriminative value was not significantly better than the separate WPI and SSS scores on which it is based. Notably, the ROC AUC of the total symptom count, which is not part of the diagnostic criteria, was also not significantly different from that of the PSD score and also the comorbid index performed comparably in identifying patients with seizures.

The PSD additionally showed to serve as an accurate surrogate measure of FM diagnosis (ROC AUC for PSD to FM diagnosis = 0.975). In the current data set, FM diagnosis began at a PSD level ≥12 and a PSD level ≥15 correctly classified 92.6% of cases. We visually explored the distribution of PSD scores in the total sample and in patients with seizures, and the effect of the PSD scale on the probability of seizures. Fig 1 shows the distribution of PSD scores in all patients in the study (upper left) and in patients with self-reported seizures (upper right). In patients with seizures, the proportion of patients with a score ≥15 was clearly much higher than in the total sample. In the lower-right panel of Fig 1, these data are further expressed by graphing the probability of self-reported seizures (y-axis) as a function of PSD levels (x-axis).

**Table 1. Prevalence and odds of seizures by fibromyalgia status in RA patients.**

| Patients | N | % | Seizure cases (N) | Prevalence (%) (95% CI) unadjusted | Prevalence (%) (95% CI) adjusted* | Odds ratio (95% CI) adjusted* |
|---|---|---|---|---|---|---|
| All | 11,378 | 100.0 | 186 | 1.63 (1.40–1.85) | 1.63 (1.40–1.87) | |
| FM (+) | 2,390 | 21.9 | 89 | 3.72 (2.97–4.42) | 3.74 (2.95–4.53) | 3.54 (2.65–4.74) |
| FM (-) | 8,988 | 79.0 | 97 | 1.08 (0.85–1.27) | 1.08 (0.87–1.30) | |

* Adjusted for age and sex. CI = confidence interval; FM = fibromyalgia.

**Table 2. ROC areas for seizure predictors in RA patients.**

| Variable | ROC area | (95% confidence intervals) |
|---|---|---|
| Polysymptomatic distress | 0.683 | (0.643–0.724) |
| Symptom severity scale* | 0.674 | (0.636–0.714) |
| Widespread pain index* | 0.661 | (0.618–0.703) |
| Symptom count* | 0.662 | (0.621–0.704) |
| Comorbidity index | 0.601 | (0.570–0.650) |

* Not significantly different from polysymptomatic distress scale.

The probability of seizures increased disproportionate to the total number of subjects as the PSD level increased. The probability of seizures was 1.6% at a PSD level of 12 and rose to 6.8% at a level of 30.

Taken as a whole, these data show that both the FM diagnostic construct and other FM-related variables were associated with seizures and that the higher the scores on these variables the more likely a patient is to report a seizure. But importantly, both those variables that are components of the FM criteria and the total symptom count and comorbidity index appeared to identify patients with seizures similarly well.

## Relation of seizures and seizure medication to FM variables and patient outcomes and characteristics

Differences in FM variables and patient-reported outcomes between RA patients with and without seizures and between seizure patients with and without seizure drugs are shown in Table 3. FM criteria were satisfied by 47.8% of patients with seizures and 20.6% of those without seizures. All other FM diagnostic and FM-related variables were markedly high in those with seizures compared with those without seizures.

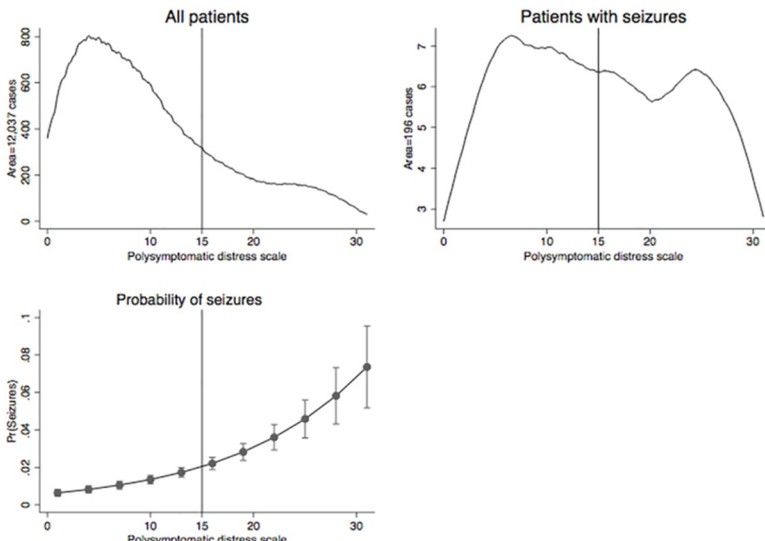

**Fig 1. Polysymptomatic distress and seizures.** Fibromyalgia positivity begins at a PSD score of ≥12 and a PSD level ≥15 correctly classified 92.6% of cases. The upper left graph shows the distribution of PSD scores in all RA patients in the study, whereas the upper right graph shows the distribution of PSD scores in patients with seizures only. The lower left graph demonstrates the direct association of PSD with seizures and shows that the probability of seizures is increasingly disproportionate to the number of subjects as the PSD level increases.

In addition, functional status (HAQ: 1.5 vs. 1.0), VAS pain (5.6 vs. 3.7) and quality of life (EQ5D: 59.7 vs. 73.1) were much worse in those with seizures. Thus, the self-report of seizures was associated with worse patient-reported outcomes. Patients which reported seizures also differed in some socio-demographic characteristic. Adjusted for age and sex, patients reporting seizures were less likely to have gone to college (28.0% vs. 38.0%, p<0.001), had lower mean household incomes ($US 37,776 vs $US 56,278, p<0.001), and were less likely to be married (62.3% vs. 70.6%, p = 0.047). They did not differ significantly from those without seizures in current smoking (12.9% vs. 11.1%, p = 0.459) or body mass index (BMI) (30.1 vs. 28.8, p = 0.057).

To further elucidate the relationship of FM variables to seizures, we characterized patients according to epilepsy treatment categories (Tables 3 and 4). As seizure treatment drugs may be used by patients without seizures, or for non-seizure purposes in those with seizures, we

**Table 3. Characteristics of seizure, non-seizure and treatment groups in RA patients.**

| | N | PSD (mean) | WPI (mean) | SSS (mean) | Generalized Pain (%) | Widespread Pain (%) | Symptom Count (mean) | FM (%) | Comorbidity Index (mean) | HAQ (mean) | Pain (mean) | EQ5D (mean) |
|---|---|---|---|---|---|---|---|---|---|---|---|---|
| (Range) | | (0–31) | (0–19) | (0–12) | (0–100) | (0–100) | (0–21) | (0–100) | (0–9) | (0–3) | (0–10) | (0–100) |
| Seizures (+) | 186 | 15.2 | 9.2 | 6.1 | 59.1 | 67.2 | 7.0 | 47.8 | 2.6 | 1.5 | 5.6 | 59.7 |
| Seizures (-) | 11,192 | 9.7 | 5.6 | 4.1 | 35.8 | 44.7 | 4.4 | 20.6 | 1.9 | 1.0 | 3.7 | 73.1 |
| All | 11,378 | 9.8 | 5.6 | 4.2 | 36.2 | 45.1 | 4.5 | 21.0 | 1.9 | 1.0 | 3.8 | 72.9 |
| Seizure drugs, seizures (+) | | | | | | | | | | | | |
| Any Seizure drug | 68 | 17.2 | 10.4 | 6.8 | 69.1 | 76.5 | 7.5 | 58.8 | 2.7 | 1.6 | 6.6 | 55.2 |
| Pure seizure drugs | 31 | 15.8 | 9.4 | 6.4 | 58.1 | 64.5 | 6.6 | 45.1 | 2.5 | 1.6 | 6.7 | 61.0 |
| Mixed seizure drugs | 14 | 17.9 | 10.4 | 7.6 | 78.6 | 85.7 | 8.1 | 71.4 | 3.4 | 1.7 | 6.6 | 49.7 |
| Mixed plus seizure drugs | 49 | 18.6 | 11.3 | 7.3 | 75.5 | 81.6 | 8.2 | 67.3 | 3.1 | 1.6 | 6.8 | 52.0 |
| Gabapentin | 37 | 19.3 | 12.0 | 7.3 | 75.7 | 81.0 | 8.0 | 67.6 | 2.9 | 1.7 | 6.9 | 49.9 |
| Seizure, but no seizure drug | 118 | 14.1 | 8.4 | 5.7 | 53.5 | 61.9 | 6.8 | 41.5 | 2.4 | 1.4 | 4.9 | 62.2 |
| Seizure drugs, all patients | | | | | | | | | | | | |
| Any Seizure drug | 1,084 | 13.6 | 8.0 | 5.6 | 53.6 | 43.9 | 6.4 | 38.6 | 2.7 | 1.4 | 5.2 | 64.0 |
| Pure seizure drugs | 81 | 13.5 | 7.6 | 5.8 | 50.6 | 60.4 | 5.8 | 37.0 | 2.7 | 1.3 | 5.2 | 65.5 |
| Mixed seizure drugs | 239 | 12.9 | 6.9 | 5.9 | 45.6 | 57.7 | 6.5 | 33.9 | 2.7 | 1.2 | 4.6 | 64.1 |
| Mixed plus seizure drugs | 1,026 | 13.7 | 7.9 | 5.8 | 54.1 | 63.9 | 6.5 | 38.9 | 2.7 | 1.4 | 5.2 | 63.9 |
| Gabapentin | 809 | 14.1 | 8.3 | 5.8 | 57.1 | 67.1 | 6.5 | 41.2 | 2.7 | 1.4 | 5.5 | 63.3 |
| Seizure, but no seizure drug | 118 | 14.1 | 8.4 | 5.7 | 53.5 | 61.9 | 6.8 | 41.5 | 2.4 | 1.4 | 4.9 | 62.2 |

Pure seizure drugs: levetiracetam, oxcarbazepine, phenobarbital, phenytoin, valproic acid, zonisamide. Mixed seizure drugs: carbamazepine, acetazolamide, clonazepam, lamotrigine, pregabalin, primidone, topiramate. Mixed plus seizure drugs: carbamazepine, acetazolamide, clonazepam, lamotrigine, pregabalin, primidone, topiramate, gabapentin. PSD = polysymptomatic distress score, WPI = widespread pain index, SSS = symptom severity scale, FM = fibromyalgia, HAQ = Health assessment questionnaire disability index, EQ5D = EuroQol, MTX = methotrexate.

**Table 4. Epileptic drugs assessed in study.**

| Epileptic drugs | Other indications | Patients Using % | Number of patients | Association with seizures Odds Ratio (C.I.) |
|---|---|---|---|---|
| All seizure drugs | | 9.5 | 1,084 | 5.8 (4.2–7.8) |
| Pure seizure drugs | | 0.7 | 81 | 44.6 (27.7–71.7) |
| Mixed seizure drugs | | 2.1 | 239 | 4.0 (2.7–6.9) |
| Mixed seizure drugs plus gabapentin | | 9.0 | 1,026 | 3.7 (2.7–5.2) |
| Acetazolamide | Glaucoma, diuretic | 0.0 | 4 | 0.1 ($1.4^{-94}$–0.4) |
| Carbamazepine | Neuropathy | 0.1 | 11 | 6.0 (0.8–47.5) |
| Clonazepam | Panic disorder, anxiety | 1.3 | 149 | 3.0 (1.4–6.6) |
| Gabapentin* | Pain | 7.1 | 809 | 3.4 (2.3–4.8) |
| Lamotrigine | Bipolar disorder | 0.3 | 28 | 10.2 (3.5–29.8) |
| Levetiracetam | | 0.3 | 30 | 99.8 (47.3–210.5) |
| Oxcarbazepine | | 0.1 | 10 | 6.7 (0.8–53.3) |
| Phenobarbital | | 0.1 | 13 | 38.6 (12.5–119.2) |
| Phenytoin | | 0.1 | 11 | 74.6 (22.6–246.6) |
| Pregabalin | Pain | 0.0 | 3 | 0.5 ($3.0^{-93}$–0.8) |
| Primidone | Tremor | 0.2 | 20 | 3.2 (.42–23.9) |
| Topiramate | Migraine | 0.4 | 49 | 5.4 (1.9–15.3) |
| Valproic acid | | 0.2 | 17 | 13.1 (3.7–45.9) |
| Zonisamide | | 0.0 | 5 | 60.0 (4.3–248.1) |

*Indicated for seizures, but uncommonly used for that purpose.

Pure seizure drugs: levetiracetam, oxcarbazepine, phenobarbital, phenytoin, valproic acid, zonisamide.

Mixed seizure drugs: carbamazepine, acetazolamide, clonazepam, lamotrigine, pregabalin, primidone, topiramate.

Mixed plus seizure drugs: carbamazepine, acetazolamide, clonazepam, lamotrigine, pregabalin, primidone, topiramate, gabapentin.

characterized drugs that are indicated for seizure treatment according to the strength of their association with seizures and then examined the characteristics of patient treatment group users in Table 3. Patients who had seizures and the users of seizure treatment drugs, with the exception of "pure" seizure drugs (drugs that tended to be used exclusively for seizure treatment), had very high levels of PSD and very high proportions of patients meeting the FM criteria (Table 3). Among patients who used "pure" seizure drugs, whether or not they reported seizures, high scores were noted compared with those without seizures. In general, patients with seizures who were using seizure treatments (not including "pure" seizure treatments) had substantial and clinically important score abnormalities. Seizure treatments, as we defined them, thus appear to be strongly associated with worse clinical status.

## Discussion

Several studies have linked fibromyalgia with seizure disorders, including epidemiological studies and clinical reports. The current study, performed using data from a large sample of RA patients from a national database and using current FM diagnostic survey criteria, confirmed a clear and significant association between self-reported seizures and both FM diagnosis and fibromyalgia-related variables. The associations of seizures with FM diagnostic criteria and other fibromyalgia-related symptoms, including the number of somatic symptoms and comorbidities, were very similar. This indicates that the diagnosis of FM itself does not add specific information explaining the observed increased prevalence of seizures, but instead suggests that seizures are associated with multi-symptom comorbidity in general.

Previous epidemiological studies examining the association between FM and seizures relied on self-reports of having a clinical diagnosis of FM. In the current study, we used the 2016 ACR fibromyalgia diagnostic criteria to define FM cases, which assures that the diagnosis was based on a standardized, comprehensive report of symptoms [24]. The current study found an age- and sex-adjusted seizure prevalence of 3.74 per 100 RA patients with FM versus 1.08 in RA patients without FM, resulting in a seizure odds ratio of 3.54 of FM+ to FM- cases. This increased association between FM and seizure is reasonably comparable to the odds ratios found in some previous epidemiological studies of FM and seizures. For instance, a recent cross-sectional study that used data of the National Epidemiologic Survey on Alcohol and Related Conditions (NESARC-III) reported a seizure odds ratio of 2.86 (95% CI 1.77 to 4.61) for participants with self-reported diagnosis of FM as compared with participants without FM in a large and representative sample (N = 36,309) of the adult US non-institutionalized population [11]. Data from the 2002 US National Health Interview Survey (NHIS) revealed a lower adjusted odds ratio of "arthritis, rheumatoid arthritis, gout, lupus, or fibromyalgia" with seizures (compared with those without seizures) of 2.3 (95% CI 1.8 to 3.4). However, in that study FM was combined with other arthritic conditions, possibly explaining the lower odds ratio [6].

In the National Population Health Survey (NPHS, N = 49,000) and the Community Health Survey (CHS, N = 130,882), which represented 98% of the Canadian population, at least 19 comorbid illnesses, including FM, were more common in those who self-reported epilepsy [2]. The FM to epilepsy prevalence ratio was 1.5 (0.9 to 2.4), and FM was found in approximately 2% of those with epilepsy. The authors concluded that people with epilepsy in the general population have a high prevalence of chronic somatic comorbid conditions, including but not limited to FM. A 2008 Epilepsy Comorbidities and Health (EPIC) survey of 172,959 respondents aged 18 or older identified self-reported epilepsy in 2% of the adult noninstitutionalized US population [4]. Respondents with self-reported epilepsy were significantly more likely than those without epilepsy to report six studied neuropsychiatric disorders (prevalence ratios 1.27 to 2.39) and 4 pain disorders (prevalence ratios 1.36 to 1.96). From the pain disorders, the association was strongest for FM (prevalence ratio = 1.96, 95% CI 1.70 to 2.25). Finally, in a study of health insurance claims and membership data from nine US health plans for the year 2012, 10,271 individuals with epilepsy were matched with 20,542 people without epilepsy [5]. Analgesic opioids were used by 26% of individuals in the group with epilepsy vs. 18% of matched controls. The group with epilepsy had a significantly higher percentage of individuals in all of 16 pain conditions examined, including FM (4% vs. 3%). The total prevalence of pain diagnosis was 51% in the group with epilepsy and 39% in the control group.

While epidemiological studies have mainly focused on diagnoses of epilepsy overall, clinical studies that mention FM have concentrated on psychogenic non-epileptic seizures (PNES). A review of PNES from a refractory epilepsy clinic over a 5 year period found 28 patients with a diagnosis of FM and 8 with a diagnosis of chronic pain [3]. After work-ups, the authors reported that 75% of these patients with FM and chronic pain taken together had PNES. No information was reported about representativeness or about how the diagnosis of FM was made. Another study identified all patients with historical diagnoses of FM by using billing diagnosis codes for an epileptologist in a neurology clinic over a three-year period [7]. From 1,730 new cases, FM was retrospectively identified in 5.5% of the patients, and women represented 95% of the FM cases. Paroxysmal events were present in 57% of FM patients. Among patients with FM and paroxysmal disorders, 11% had epileptic seizures, 74% had psychogenic non-epileptic seizures, and 15% had physiological non-epileptic events. The authors concluded that in patients with undifferentiated paroxysmal events, a history of FM had a positive

predictive value of 74% for the diagnosis of PNES, which was similar to previous retrospective studies.

Results of such studies tend to be cited in other reports, leading to a perception that FM is associated with seizures in general, and specifically with PNES. Certain additional conclusions can be drawn from those cited studies: in population studies, persons with self-reported seizure disorders have an increased prevalence of many medical conditions and many different symptoms, and self-reported FM is one of those conditions. In general, the point prevalence of FM in population studies is approximately 2 times greater in those with seizures compared to those without seizures [4, 6, 11]. When it comes to clinical studies, the nature of the clinical setting causes the focus to be more on PNES, as gaining adequate control subjects in non-population clinical studies is very difficult. Clinical studies seem to show that ~75% of persons with FM can be found to have PNES [4, 7]. There are, however, a number of difficulties with the above reports. No epidemiological study identified FM using valid diagnostic criteria. Instead either self-report of clinical diagnosis or billing diagnosis were used to identify persons as having FM. As we discus below, it is likely that the accuracy of the diagnosis in these studies was low. In addition, clinical studies noted above had severe referral and selection biases which limits generalizability. This is not to fault the authors of these epidemiological studies because no other data were available, nor the authors of the clinical studies since they reported on what was present in the clinic. It is likely that the observed existence of an association of seizures with FM in these studies is correct, but that the strength of this association may not be accurate.

Much of the work of the authors of the current study has involved the definition, prevalence, clinical characteristics and validity of the concept and diagnosis of FM. From that perspective, it is important to offer some comments on the current status of FM that may be helpful for those involved in seizure diagnosis and care. Fibromyalgia is a constructed disorder that provides a diagnostic name for symptoms experienced by people who self-report widespread pain, high levels of somatic symptoms, fatigue, substantial difficulties with sleep, and cognition problems ("fogginess"). There is no gold standard for FM, and its definition has been established and revised several times by committees [1, 22, 24, 37]. Although there are both research and clinical criteria for FM, 75% of people in populations studies who report a physician's diagnosis of FM do not satisfy the current criteria for the disorder, and an equal number would satisfy criteria but have not been given the diagnosis [17]. In addition, there is evidence that most people diagnosed with FM in clinical practice are given the diagnosis based on gestalt that includes behavioral, social and symptom characteristics.

"No one has fibromyalgia until it is diagnosed" [13], but the diagnosis is often arbitrary. Physicians are not required to diagnose FM. For example, patients with low back and leg pain who would satisfy FM criteria, may instead receive a diagnosis of low back pain. If FM is primarily identified on the basis of behavioral, social and symptom characteristics rather than published criteria, then the patients so diagnosed are likely to be women, and to have more psychological and symptom complaints [18]. In contrast to clinical studies (including seizure studies) that find 95% of those with FM to be women, population studies with modern criteria find percentages around 60% to be women [38, 39]. If physicians suspect FM in women but not in men, and they use gestalt criteria, then women will be over-diagnosed, men will be under-diagnosed, and psychological symptoms will be emphasized [18].

In the current study, we found that FM criteria positivity in RA patients was indeed significantly associated with a more than threefold increased odds of seizures. Noteworthy, all of the components that make up FM as defined by the criteria (PSD, SSS, and WPI) were similarly associated with increased probability of seizures. However, the discriminative values for number of comorbid conditions and the total symptom counts, which are not part of the diagnostic

criteria, were also of similar magnitude. These findings, and the fact that the variables occur together, show that various symptoms, symptom counts [8, 31], FM diagnosis and chronic pain are similarly associated with seizures. Thus, there appears to be nothing unique about FM as a separate diagnosis with respect to seizures and seizure disorders.

There is no clear direction of causality with respect to the association between fibromyalgia variables and seizures, a conclusion that also has been reached by Keezer [10]. It may be that, in patients with seizures, PSD and fibromyalgia, as defined in the present study, are the result of physical and mental stressors or distress. It is known that individuals with seizures undergo much stress in their everyday lives [40–43].

There are a number of limitations to our study, most importantly the reliance on self-reported seizures. Although many epidemiologic studies also relied on self-report of seizures [2, 4–6, 11], this does limit both the clinical and biological interpretation of the associations found between FM and seizures. First of all, as we did not use the word "epilepsy" in our questionnaire, it remains unclear to what extend the self-reported seizures of the patients are based on epilepsy or PNES. There is a high likelihood that most of the seizures we identified are psychogenic, but this cannot be (clinically) verified using self-reported data, such as those in the NDB data and population studies in general. Although similar in their external manifestations (and sometimes coexisting in the same patient [44]), however, epileptic and psychogenic seizures are considered to have completely different etiologies. Second, patients may use the term seizures informally for any type of episodes of loss of awareness or focal neurological deficit, including stroke, syncope or other diagnosis. This might have led to both increased reporting and confounded associations with FM Even though we also examined the use of dedicated antiepileptic medication, this is no guarantee for the presence of epilepsy as some of those medications are also prescribed to FM patients. Therefore, future studies should use both validated FM criteria assessments and clinically verified diagnoses of epileptic and psychogenic seizures in order to more rigorously assess the association between FM and seizures and to be able to distinguish associations with epileptic and psychogenic nonepileptic seizures.

For evaluation of the relation between FM and seizures, our study used patients with rheumatoid arthritis. The use of rheumatoid arthritis patients has several advantages, in particular unbiased selection, as patients were not specifically selected for characteristics related to seizures or FM. In addition, using a rheumatoid arthritis population provides extensive clinical data and enrichment of the study by increased number of fibromyalgia patients. On the other hand, previous studies have indicated that autoimmune disorders in general, and RA in specific, are by themselves also associated with increased risk of seizures, which may be explained by vasculitis, central nervous system infections and the use of methotrexate and sulphasalazine [45]. Therefore, the associations demonstrated in this study may not hold in the general population. A recent meta-analysis of six nationwide population-based studies showed a pooled epilepsy prevalence of 0.83% in RA patients versus 0.44% in non-RA patients, resulting in a risk ratio for epilepsy of 1.601 (95% CI 1.089 to 2.354) in RA vs. non-RA [46]. Consequently, the seizure odds ratio of 3.54 for FM+ patients found in the current study may be an underestimation of the actual association between FM diagnosis and seizures in general non-RA populations, but this speculation needs to be confirmed in general population studies.

## Conclusions

In summary, we found a significant association of FM diagnostic and FM-related variables, including the number of somatic symptoms and comorbidities, with self-reported seizures among people with RA. This association is similar to that found in previous studies about symptom variables and seizures and does not suggest any special role for FM diagnosis. It

suggests that it is rather multi-symptom comorbidity that is linked to seizures, and this would be in a complex and not clearly understood way. RA patients reporting seizures had more pain, more functional limitations and a reduced quality of life. Seizures patients treated with anti-seizure medication had generally worse outcomes than those without.

## Supporting information

**S1 Data.**
(DTA)

**S1 Text.**
(TXT)

## Author Contributions

**Conceptualization:** Johannes J. Rasker, Frederick Wolfe.

**Data curation:** Frederick Wolfe.

**Formal analysis:** Frederick Wolfe.

**Resources:** Frederick Wolfe.

**Writing – original draft:** Johannes J. Rasker, Peter M. ten Klooster.

**Writing – review & editing:** Frederick Wolfe, Ewa G. Klaver-Krol, Machiel J. Zwarts.

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
