## [Decision Letter · Decision Letter 0]

2 Nov 2020

PONE-D-20-29759

The association of fibromyalgia and fibromyalgia-related symptoms with self-reported seizures

PLOS ONE

Dear Dr. Klooster:

Thank you for submitting your manuscript to PLOS ONE. After careful consideration, we feel that it has merit but does not fully meet PLOS ONE’s publication criteria as it currently stands. Therefore, we invite you to submit a revised version of the manuscript that addresses the points raised during the review process.

We look forward to receiving your revised manuscript.

Kind regards,

Hyunmi Choi, MD, MS

Academic Editor

PLOS ONE

Additional Editor Comments:

Dear Dr. Klooster:

Your paper has been reviewed by our reviewers. Enclosed are the reviewers' comments, which I urge you to address in revising your manuscript. Please consider Reviewer 1's comments about the use of self-reported seizures, and further expand your limitation section in Discussion. Also, please include a sentence in the Abstract to balance your conclusion, with a statement about the limitations of using self-reported seizure, specifically about possible inclusion of those with psychogenic nonepileptic seizures. Please make the recommended changes as per Reviewer 2.

Sincerely,

Hyunmi Choi, MD, MS

Academic Editor

Journal Requirements:

2. Please provide additional details regarding participant consent. In the ethics statement in the Methods and online submission information, please ensure that you have specified what type you obtained (for instance, written or verbal, and if verbal, how it was documented and witnessed).

3. Please include additional information regarding the questionnaires used in the study and ensure that you have provided sufficient details that others could replicate the analyses. For instance, if you developed the questionnaire(s) as part of this study and it is not under a copyright more restrictive than CC-BY, please include a copy, in both the original language and English, as Supporting Information. If the questionnaire is published, please provide a citation.

In addition, please provide the 60 symptom checklist as a supplementary file.

Reviewers' comments:

Reviewer's Responses to Questions

**Comments to the Author**

1. Is the manuscript technically sound, and do the data support the conclusions?

Reviewer #1: Partly

Reviewer #2: Partly

2. Has the statistical analysis been performed appropriately and rigorously? 

Reviewer #1: Yes

Reviewer #2: Yes

3. Have the authors made all data underlying the findings in their manuscript fully available?

Reviewer #1: Yes

Reviewer #2: Yes

4. Is the manuscript presented in an intelligible fashion and written in standard English?

Reviewer #1: Yes

Reviewer #2: Yes

5. Review Comments to the Author

Reviewer #1: Summary:

This studies intends to describe the association of fibromyalgia with self reported seizures. Overall study is well written

Major comments:

- This paper has a very fundamental issue, which is that the definition of seizures used out of patient self report is not a rigorous one. Patients often use the term seizures very informally for any type of episodes of loss of awareness or focal neurological deficit. That may include stroke, epilepsy, syncope and other diagnosis. So using patient self report without verification by a physician can just answer the question does fibromyalgia has association with any neurological symptom. Second there is the problem that typically neurologists refer to seizures as repetitive neurological disturbance caused by electrical brain disturbance. So psychogenic events (PNES) is not typically considered a seizure. So no sure that this paper will provide any useful information for clinical practice.

- Perhaps the medication criteria is a better surrogate of possible diagnosis of seizures/epilepsy even though not perfect. However, why where these list of Antiepileptic medications were chosen ? our of convinience or what it was available in the database. There is more anticonvulsants in existence not in the list and very common like valproic acid (depakote), levetiracetam etc.

- I dont know about the value of the Seizure drug analysis. Furthermore, the division of the drugs into Pure drugs, mixed and mixed plus makes the study overly complicated and provide very little or limited clinical value. I do understand that authors are trying to compensate for the seizure self reporting nature of the study. Why not rather use the drugs as a way to assess accuracy of possible epilepsy diagnosis if possible.

- The point made in line 290 further confirm the concern about the use of self reporting seizure. Some of the patient using "pure" seizure drug have more likelyhood of FM even when they report no seizures. Now why would that be the case. Normally people who take those drugs should report seizures. Not sure the dataset is robust enough to be confident on the seizures. THe FM diagnosis seems fairly robust but not sure seizure are

Minor comments:

- It would be useful to the unfamiliar reader with the dataset if the author describe the response rate by patients for this questionnaire and how rigorous is the questionnaire administration process.

- Results Line 214: I will add % of seizures as it is not intuitive that 89 FM+ report sz an 97 FM- but 89 represented higher rate of seizures within that group.

Reviewer #2: The authors examine the association between fibromyalgia (assessed using validated self-reported symptom-based diagnostic criteria) and self-reported seizures, among patients with rheumatoid arthritis, using the National Databank for Rheumatic Diseases. This study follows multiple clinical and epidemiological studies that have previously examined this association, with similar findings. They conclude that there is a significant association between fibromyalgia (as well as multi-symptom comorbidities more generally) and self-reported seizures.

The study is overall clinically valuable in that it confirms, strengthens, and somewhat elucidates prior findings of an association between fibromyalgia and seizures. The methodology is generally sound, and the writing is generally clear and precise.

This study does make an important improvement in that it assesses fibromyalgia based on validated criteria rather than self-reported diagnosis or unstandardized clinical diagnosis. There is substantial reason to think that clinical fibromyalgia diagnoses are often inaccurate, so this is a notable improvement. However, unlike some prior clinical studies, the current study assesses seizures based solely on self-report, and does not differentiate between epileptic or psychogenic seizures. Epileptic and psychogenic seizures are thought to have entirely different etiologies, so an association with self-reported undifferentiated “seizures” is of unclear clinical or biological significance. As the authors note, multiple prior clinical studies have demonstrated a strong association between self-reported fibromyalgia and psychogenic seizures specifically, so readers might well wonder what fraction of the self-reported seizures in this study are psychogenic.

The authors appropriately note both the reliance on self-report of seizures and the failure to differentiate between epileptic and psychogenic seizures in the limitations. I think they might go further and suggest that future studies might use both validated assessments of fibromyalgia and, at the very least, clinical diagnoses of epileptic and psychogenic seizures.

This study is also importantly limited in that it is conducted entirely in a population with rheumatoid arthritis. The authors present this solely as a strength of the study, noting that patients with rheumatoid arthritis have higher rates of fibromyalgia, and also higher rates of seizures. But this is also a significant limitation in that the associations demonstrated in this study may not hold in the general population. The authors speculate that the association may be even stronger in the general population, but the study offers no actual data regarding the association between fibromyalgia and seizures in the general population. This fact is not reflected in the study’s conclusions which state “we found a significant association of FM diagnostic and FM-related variables including the number of somatic symptoms and comorbidities, with self-reported seizures” (lines 436-437). The conclusions as currently written are not supported by the evidence in the study. The concluding sentence should end with a caveat such as “among people with RA.” Similar caveats should be added in the abstract.

The authors also evaluate seizure medications among patients with and without seizures, demonstrating that seizure patients treated with anti-seizure medications had worse outcomes than those not so treated. As the authors note, the presence or absence of anti-seizure medications is not a good marker for epileptic vs. psychogenic seizures. The authors appropriately break these medications down into those that are usually used to treat seizures and those that are commonly used for other purposes. Surprisingly they do not include levetiracetam at all, which is today among the most commonly prescribed seizure medications. Also surprisingly, primidone is listed as a pure seizure drug when in adults it is more commonly prescribed for tremor. Acetazolamide, which is listed as being used for glaucoma and diuresis, is also commonly used for idiopathic intracranial hypertension. I would suggest including levetiracetam if at all possible and switching primidone from the pure to the mixed seizure drug category.

The manuscript includes a number of typos and grammatical errors, and would benefit from close line editing. For example:

Lines 153-154: “The continues PSD score has been shown to…”

Lines 162-163: “A total symptom count was calculated for this study based on a checklist of 60 specific symptom, dived over 8 categories…”

6. PLOS authors have the option to publish the peer review history of their article (what does this mean?). If published, this will include your full peer review and any attached files.

Reviewer #1: No

Reviewer #2: **Yes: **Benjamin Tolchin, MD, MS

---

## [Author Response · Author response to Decision Letter 0]

17 Dec 2020

Point-by-point reply

Additional Editor Comments:

Your paper has been reviewed by our reviewers. Enclosed are the reviewers' comments, which I urge you to address in revising your manuscript. Please consider Reviewer 1's comments about the use of self-reported seizures, and further expand your limitation section in Discussion. Also, please include a sentence in the Abstract to balance your conclusion, with a statement about the limitations of using self-reported seizure, specifically about possible inclusion of those with psychogenic nonepileptic seizures. Please make the recommended changes as per Reviewer 2.

Reply: We have incorporated the reviewer 1’s comments about the use of self-reported seizures, expanded the section on this limitation in the discussion and added this limitation to the abstract. The recommended changes by reviewer 2 (regarding medications) were made and we re-ran the analyses accordingly, which resulted in some changes in Tables 3 and 4, but which did not change the conclusions. 

We also slightly modified the title of the manuscript from “The association of fibromyalgia and fibromyalgia-related symptoms with self -reported seizures” to “The relation of fibromyalgia and fibromyalgia-related symptoms to self -reported seizures” as we think this better fits with the study goal. 

Journal Requirements:

Reply: We have checked all style requirements. 

2. Please provide additional details regarding participant consent. In the ethics statement in the Methods and online submission information, please ensure that you have specified what type you obtained (for instance, written or verbal, and if verbal, how it was documented and witnessed).

 Reply: Participants in the NDB are volunteers who agree to complete questionnaires at 6-month intervals. Each patient completes a written consent that is part of the larger 27 page questionnaire. 

 This information was added to the ethics statement (page 9).

3. Please include additional information regarding the questionnaires used in the study and ensure that you have provided sufficient details that others could replicate the analyses. For instance, if you developed the questionnaire(s) as part of this study and it is not under a copyright more restrictive than CC-BY, please include a copy, in both the original language and English, as Supporting Information. If the questionnaire is published, please provide a citation.

In addition, please provide the 60 symptom checklist as a supplementary file.

 Reply: The questionnaire that is germane to this study is a 27-page questionnaire that has been used in hundreds of papers. The full survey can be downloaded from: https://www.arthritis-research.org. The 60 symptom checklist is part of this questionnaire.

 This information has been added to the manuscript (page 6).

Reviewer #1:

This study intends to describe the association of fibromyalgia with self reported seizures. Overall study is well written.

Major comments:

This paper has a very fundamental issue, which is that the definition of seizures used out of patient self report is not a rigorous one. Patients often use the term seizures very informally for any type of episodes of loss of awareness or focal neurological deficit. That may include stroke, epilepsy, syncope and other diagnosis. So using patient self report without verification by a physician can just answer the question does fibromyalgia has association with any neurological symptom. Second there is the problem that typically neurologists refer to seizures as repetitive neurological disturbance caused by electrical brain disturbance. So psychogenic events (PNES) is not typically considered a seizure. So no sure that this paper will provide any useful information for clinical practice.

Reply: Certainly, we are aware of the problem. In the paper we distinguish 3 types of seizure classifications: 1) Self-reported seizures, 2) PNES and 3) epileptic seizures. As the reviewer points out, and we make clear in the paper, we only have data on self-reported seizures. The reason why we think such data are still valuable is that many population surveys as well as neurological papers address self-reported seizures. In addition, PNES (self-reported) seizures are the widely seen type of seizure seen by neurologists according to the literature – including written links to fibromyalgia. Thus, the data we provide has useful information about this large and frequently reported group. 

The reviewer might also wonder why we don’t report on epileptic seizures. The answer is that it is impossible to obtain reliable data on this point. We (NDB) can obtain hospital records easily enough (and we do), but communications with and verification by physicians about seizures is virtually impossible.

Nonetheless, we certainly acknowledge this fundamental problem and have expanded the section on this in the Discussion (page 17) and added this as a limitation in the abstract. 

Perhaps the medication criteria is a better surrogate of possible diagnosis of seizures/epilepsy even though not perfect. However, why where these list of Antiepileptic medications were chosen ? our of convenience or what it was available in the database. There is more anticonvulsants in existence not in the list and very common like valproic acid (depakote), levetiracetam etc.

Reply: First, we want to point out an error that we made. We inadvertently omitted valproic acid for our analyses, although we had data on this treatment. We also did not include levetiracetam because of a coding error (indicating no use).These errors have been corrected. The reviewer is correct that there was a certain arbitrariness in our list of drugs because we sought to not have a long list of drugs that were not used. However, the reviewer’s remarks show that we did not enumerate the drugs correctly. 

To correct this issue, we have done following. We obtained a list of 29 anti-seizure drugs using data from the Epilepsy Foundation (https://www.epilepsy.com/learn/treating-seizures-and-epilepsy/seizure-medication-list). In Table 3, we now report results from the drugs on this list that were used by patients in the study. This results in the addition of valproic acid and levetiracetam to Table 3 and the removal of several drugs from that table that were actually not used by study patients. Furthermore, this addition of valproic acid and levetiracetam results in very slight and unimportant changes to the results of Tables 3 and 4.

I don’t know about the value of the Seizure drug analysis. Furthermore, the division of the drugs into Pure drugs, mixed and mixed plus makes the study overly complicated and provide very little or limited clinical value. I do understand that authors are trying to compensate for the seizure self reporting nature of the study. Why not rather use the drugs as a way to assess accuracy of possible epilepsy diagnosis if possible.

Reply: With respect to “Why not rather use the drugs as a way to assess accuracy of possible epilepsy diagnosis if possible,” we have no way really to tell the accuracy of the diagnosis as it is very likely that many patients with what would be PNES are treated with anti-epileptic drugs. Rather, the data show a kind of agreement between patients and prescribers even though such diagnosis may be incorrect. The data are presented to outline the conundrum of diagnosis and treatment when there is uncertainty about PNES/epileptic status. 

The point made in line 290 further confirm the concern about the use of self reporting seizure. Some of the patient using "pure" seizure drug have more likelihood of FM even when they report no seizures. Now why would that be the case. Normally people who take those drugs should report seizures. Not sure the dataset is robust enough to be confident on the seizures. The FM diagnosis seems fairly robust but not sure seizure are

Reply: With respect to: “Some of the patient using "pure" seizure drug have more likelihood of FM even when they report no seizures. Now why would that be the case. Normally people who take those drugs should report seizures.” We think the answer is that patients who did not have seizure while on medications during the last 6 months may simply have had seizures in the past.

Minor comments:

It would be useful to the unfamiliar reader with the dataset if the author describe the response rate by patients for this questionnaire and how rigorous is the questionnaire administration process.

Reply. We have reported previously on rates of missing data. One of our papers states, “All participants completed at least once a comprehensive 28-page questionnaire that included all study questions. Over the course of the study the comprehensive questionnaire was completed at 92.7% of observations, and had missing data rates for HAQ, PCS, and MCS of 0.4%, 3.2%, and 3.2%, respectively …” We have added the following to the text (page 6): 

Data capture rates in the NDB are high. In general, our comprehensive questionnaires and electronic equivalents have been completed at 92.7% of observations, and had missing data rates for common variables such as the Health Assessment Questionnaire (HAQ) and SF-36 of 0.4% and 3.2%, respectively [27].

Results Line 214: I will add % of seizures as it is not intuitive that 89 FM+ report sz an 97 FM- but 89 represented higher rate of seizures within that group.

Reply. We have made this correction (page 10). 

Reviewer #2:

The authors examine the association between fibromyalgia (assessed using validated self-reported symptom-based diagnostic criteria) and self-reported seizures, among patients with rheumatoid arthritis, using the National Databank for Rheumatic Diseases. This study follows multiple clinical and epidemiological studies that have previously examined this association, with similar findings. They conclude that there is a significant association between fibromyalgia (as well as multi-symptom comorbidities more generally) and self-reported seizures.

The study is overall clinically valuable in that it confirms, strengthens, and somewhat elucidates prior findings of an association between fibromyalgia and seizures. The methodology is generally sound, and the writing is generally clear and precise.

This study does make an important improvement in that it assesses fibromyalgia based on validated criteria rather than self-reported diagnosis or unstandardized clinical diagnosis. There is substantial reason to think that clinical fibromyalgia diagnoses are often inaccurate, so this is a notable improvement. However, unlike some prior clinical studies, the current study assesses seizures based solely on self-report, and does not differentiate between epileptic or psychogenic seizures. Epileptic and psychogenic seizures are thought to have entirely different etiologies, so an association with self-reported undifferentiated “seizures” is of unclear clinical or biological significance. As the authors note, multiple prior clinical studies have demonstrated a strong association between self-reported fibromyalgia and psychogenic seizures specifically, so readers might well wonder what fraction of the self-reported seizures in this study are psychogenic.

Reply: We, of course, agree with the reviewer’s point. We have extended the text on this limitation in the Discussion section (page 17).

The authors appropriately note both the reliance on self-report of seizures and the failure to differentiate between epileptic and psychogenic seizures in the limitations. I think they might go further and suggest that future studies might use both validated assessments of fibromyalgia and, at the very least, clinical diagnoses of epileptic and psychogenic seizures.

Reply: As suggested by the reviewer, we added this to both the abstract and the discussion (page 17). 

This study is also importantly limited in that it is conducted entirely in a population with rheumatoid arthritis. The authors present this solely as a strength of the study, noting that patients with rheumatoid arthritis have higher rates of fibromyalgia, and also higher rates of seizures. But this is also a significant limitation in that the associations demonstrated in this study may not hold in the general population. The authors speculate that the association may be even stronger in the general population, but the study offers no actual data regarding the association between fibromyalgia and seizures in the general population. This fact is not reflected in the study’s conclusions which state “we found a significant association of FM diagnostic and FM-related variables including the number of somatic symptoms and comorbidities, with self-reported seizures” (lines 436-437). The conclusions as currently written are not supported by the evidence in the study. The concluding sentence should end with a caveat such as “among people with RA.” Similar caveats should be added in the abstract and in the discussion (page 18).

Reply: We have added cautions as the reviewer suggest in both the abstract and conclusion sections. 

The authors also evaluate seizure medications among patients with and without seizures, demonstrating that seizure patients treated with anti-seizure medications had worse outcomes than those not so treated. As the authors note, the presence or absence of anti-seizure medications is not a good marker for epileptic vs. psychogenic seizures. The authors appropriately break these medications down into those that are usually used to treat seizures and those that are commonly used for other purposes. Surprisingly they do not include levetiracetam at all, which is today among the most commonly prescribed seizure medications. Also surprisingly, primidone is listed as a pure seizure drug when in adults it is more commonly prescribed for tremor. Acetazolamide, which is listed as being used for glaucoma and diuresis, is also commonly used for idiopathic intracranial hypertension. I would suggest including levetiracetam if at all possible and switching primidone from the pure to the mixed seizure drug category.

Reply: “I would suggest including levetiracetam if at all possible and switching primidone from the pure to the mixed seizure drug category.” We have now done this, as the reviewer suggested. Our omission was based on a coding error which is now fixed.

As we also noted in our reply to reviewer 1, we made two errors in our seizure drug analysis. We inadvertently omitted valproic acid. Furthermore, we had a coding error in the naming of levetiracetam which led to our omitting it. This has now been corrected. We have also made the mixed seizure drug category change that the reviewer suggested. Al this has resulted in some slight changes to the results of Tables 3 and 4, but did not change any of the conclusions.

The manuscript includes a number of typos and grammatical errors, and would benefit from close line editing. For example:

Lines 153-154: “The continues PSD score has been shown to…”

Lines 162-163: “A total symptom count was calculated for this study based on a checklist of 60 specific symptom, dived over 8 categories…”

Reply: We thank the reviewer for pointing out this typos which have been corrected. The entire manuscript was checked for additional typos and grammatical errors.

---

## [Editor Report · Decision Letter 1]

13 Jan 2021

The relation of fibromyalgia and fibromyalgia symptoms to self-reported seizures

PONE-D-20-29759R1

Dear Dr. Klooster,

We’re pleased to inform you that your manuscript has been judged scientifically suitable for publication and will be formally accepted for publication once it meets all outstanding technical requirements.

Kind regards,

Hyunmi Choi, MD, MS

Academic Editor

PLOS ONE

Additional Editor Comments (optional):

I appreciate the authors' responses to reviewers' comments.

---

## [Editor Report · Acceptance letter]

26 Jan 2021

PONE-D-20-29759R1 

The relation of fibromyalgia and fibromyalgia symptoms to self-reported seizures 

Dear Dr. ten Klooster:

I'm pleased to inform you that your manuscript has been deemed suitable for publication in PLOS ONE. Congratulations! Your manuscript is now with our production department. 

Kind regards, 

on behalf of

Dr. Hyunmi Choi 

Academic Editor

PLOS ONE